# A Comparative Analysis of Social Entrepreneurship and Entrepreneurship: An Examination of International Co-Authorship Networks

**Karin Kurata** [1], **Shuto Miyashita** [2], **Shintaro Sengoku** [2], **Kota Kodama** [3] **and Yeong Joo Lim** [4,*]

1   Course of Information Systems Engineering, National Institute of Technology, Tsuruoka College, 104 Sawada, Inooka, Tsuruoka-city 997-8511, Yamagata, Japan; k.kurata@tsuruoka-nct.ac.jp
2   School of Environment and Society, Tokyo Institute of Technology, 2-12-1 Okayama, Meguro, Tokyo 152-8550, Japan; miyashita.s.ae@m.titech.ac.jp (S.M.); sengoku.s.aa@m.titech.ac.jp (S.S.)
3   School of Data Science, Nagoya City University, 1 Yamanohata, Mizuho-cho, Mizuho-ku, Nagoya 467-8601, Aichi, Japan; kkodama@fc.ritsumei.ac.jp
4   College of Business Administration, Ritsumeikan University, 2-150 Iwakura-cho, Ibaraki-city 567-8570, Osaka, Japan
*   Correspondence: lim40@fc.ritsumei.ac.jp; Tel.: +81-0726652393

**Abstract:** This study aimed to identify the boundaries between social entrepreneurship and entrepreneurship research through conducting a comparative analysis of international co-authorship networks. Analyzing 29,510 papers published in the Web of Science database from 1999 to 2021, this study utilized bibliometric analysis to examine international co-authorship networks, the strength of international co-authorship, and the top collaborative and collaborating countries. The results found that based on quantitative analysis, social entrepreneurship research focuses more on local challenges and less on international collaboration as compared to entrepreneurship research. Moreover, the findings reveal the involvement of developed countries in the international co-authorship for social entrepreneurship research field. This study sheds light on the characteristics of social entrepreneurship research, which focuses on local and regional challenges. Contrastingly, entrepreneurship research focuses on the globalized field while sharing information and technology. These insights could benefit researchers, practitioners, and educators in prioritizing globalization in entrepreneurship and localization in social entrepreneurship.

**Keywords:** social entrepreneurship; entrepreneurship; bibliometric analysis; social network analysis; co-authorship analysis

## 1. Introduction

"Social entrepreneurship is the field in which entrepreneurs tailor their activities to be directly tied with the ultimate goal of creating social value" [1]. In recent years, there has been growing interest among researchers in the literature on social entrepreneurship [2–16]. Unlike traditional entrepreneurship, social entrepreneurs prioritize creating social value rather than monetary benefits [17–20]. Social entrepreneurship aims to address societal challenges and meet fundamental human needs through innovative resource integration. For instance, microcredit organizations (e.g., Grameen Bank) provide people with insufficient funds the working capital needed to start a business [9]. Moreover, Sekem in Egypt shares similar features with Grameen Bank in terms of creatively mobilizing resources which they do not own. Sekem was founded by Dr. Ibrahim Abouleish in 1977 [21]. Sekem has reduced pesticide use in Egyptian cotton fields by 90% and created institutions such as schools, universities, adult educational centers, and medical centers [21]. Based on the discussion above, this provides an example of how social entrepreneurship takes social challenges and mitigates them through developing social and economic value [21]. Both social entrepreneurship and entrepreneurship have been extensively explored by researchers,

yet the boundaries between them are a controversial topic [3,21–25]. The definition of entrepreneurship is fragmented [3], and since social entrepreneurship is still in the early stage of development, this same fragmentation is evident in the literature. Furthermore, the social entrepreneurship and entrepreneurship research fields share a close relationship. Economic value is inseparable from social value [22], which poses challenges for researchers in differentiating and identifying the similarities and differences between them. To mitigate this challenge, previous studies have conducted bibliometric analysis to investigate the research fields of social entrepreneurship and entrepreneurship separately [12,26–32]. A previous study found the most influential publication using bibliographic coupling analysis and developed its network. Bibliographic coupling can identify the number of citations that two publications share in their reference lists [12]. Another study conducted co-citation analysis on entrepreneurship publications to identify the relationships between key ideas in a scientific publication. The results identified highly cited publications and the chronological flow of theory in the field of entrepreneurship research from the years 1962 to 2013 [31]. Building on these works, the current study will highlight regional aspects to specifically explain how the international co-authorship structures of these two research fields are different and similar to each other.

## 2. Literature Review

Table 1 presents the definition of social entrepreneurship. A previous study identified that one of the aims of social entrepreneurship is to act as a catalyst for social change [17]. Also, a previous study stated that creating social value is the main trait distinguishing social entrepreneurship from entrepreneurship [12]. Moreover, social entrepreneurship is also known to share characteristics with not-for-profit organizations. This is also implied for-profit organizations with a social mission or hybrid organizations which mix social and entrepreneurial practices and objectives [33]. From a practical point of view, researchers have been focusing on the aspects of the combination and mobilization of resources [34], including approaches to accessing and utilizing resources [35]. The discussion above demonstrates that social entrepreneurship shares an underlying drive for social entrepreneurship to create social value rather than personal and shareholder wealth, and related activity is characterized as innovation or the creation of something new rather than the replication of existing practices [2]. The current study will define social entrepreneurship as "a set of creative and effective activities, focusing strategically on addressing social market failure and creating new opportunities for systematic increase in social value" [12]. Table 2 presents a definition of entrepreneurship. Entrepreneurship can be defined as the development of productivity [36], pursuit of opportunity, business creation, uncertainty, and profit seeking [37]. Similarly, commercial entrepreneurship pursues the creation of profit for economic growth, which results in the creation of wealth and private gain [2,38]. In this study, we consider commercial entrepreneurship as a subset of entrepreneurship, without separating these two concepts.

**Table 1.** Definition of social entrepreneurship.

| Social Entrepreneurship | |
| --- | --- |
| Catalyst for social change | [2,12,17,21,36,39,40] |
| Social value development | [12,21] |
| Characteristic as non-profit organization (NPO) | [17,33,41] |
| Mitigate social challenge by mobilizing resources | [16,33–35] |

**Table 2.** Definition of entrepreneurship.

| Entrepreneurship | |
| --- | --- |
| Development of productivity | [21,36] |
| Commercial activities | [22,36,42] |
| Development of economic value | [15,17,21,22,43] |

## 2.1. Similarities and Differences between Social Entrepreneurship and Entrepreneurship

Both social entrepreneurship and entrepreneurship aim to create social and economic value. However, social entrepreneurship specifically focuses on the generation of social value [1,12,44–46]. Moreover, while social entrepreneurship emerged as a subfield of entrepreneurship [47], they share some similarities. However, entrepreneurship primarily focuses on stakeholders and access to financial opportunities provided by private investors, which can be more challenging for social entrepreneurship [36]. Social entrepreneurship aims to remain financially self-sufficient [1] due to the lack of access to financial resources. Entrepreneurship is attractive to investors such as the government for creating employment opportunities, developing productive growth, and delivering high-quality commercialization [36]. Consequently, social entrepreneurship often faces difficulties in accessing the same capital markets as commercially oriented entrepreneurship since performance measurement with financial indicators, among other measures, is rarely available [2]. Previous studies have explored the definitional and conceptual aspects of social entrepreneurship [15,48] regarding comparative analyses between social entrepreneurship and entrepreneurship in educational [49], theory [2,50], and financial risk management [51]. Against this background, we address two major conceptual limitations.

## 2.2. Conceptual Aspects of Social Entrepreneurship and Entrepreneurship

Previous studies have provided initial insights into social entrepreneurship [12,26–30], entrepreneurship [31], and the entrepreneurship ecosystem [32] using bibliometric analysis. These studies have identified the most influential authors and journals [12] through conducting co-authorship [26,52] and international co-authorship analyses [26,28]. For instance, a previous study that conducted bibliometric analysis on institutional collaboration in the social entrepreneurship research field found that almost half of the cases identified took place in England from the years 2005 to 2017 [26]. A previous study also conducted a study on chronological change in the theory of entrepreneurship through co-citation analysis. One study found that six different theories supporting and changing the scientific structure of entrepreneurship have been introduced from 1962 to 2013 [27]. A previous study discussed the significance of conducting comparative analysis while utilizing bibliometric analysis to study tourism and hospitality [53]. This study found that co-citation and co-author analyses are useful in revealing the relationships between scholar; however, they limit the contribution of identifying the relationships between different fields and areas [53]. Therefore, it is important to understand the significance of the research field of social entrepreneurship within the broader field of entrepreneurship. However, no study has quantitatively analyzed both research fields simultaneously. This study advances our understanding by identifying the characteristics of entrepreneurship and social entrepreneurship.

## 2.3. Geographic Aspects of Social Entrepreneurship and Entrepreneurship

Comparative analyses have revealed that social entrepreneurship faces financial challenges, since financial incentives are rarely available [2], given that its aim is to be financially independent [1]. Moreover, social entrepreneurship is established through the involvement of community and volunteers [2]. Robust networks are critical in allowing social entrepreneurs to gain resources including funding, staff, and so on [2]. In doing so, trust, reputation, and skill in dealing with key players' needs are important [2]. Therefore, social entrepreneurship must consider the needs and challenges specific to local contexts, which can differ from the approach adopted by entrepreneurial activities. For instance, many social innovations have been created in a locally embedded context [3]. Based on the discussion above, it is implied that previous studies considered the regional aspects of entrepreneurship [54–56] and social entrepreneurship [57,58] separately. For instance, a previous study found that the configuration, efficiency, and sustainability of a regional ecosystem that improves economic development through entrepreneurship is the result of actors in a specific location [55]. Moreover, a previous study analyzed how geographic area

influences the types of social networks in which social entrepreneurship is embedded [58]. They found that social entrepreneurs who seek more embedded community relationships are likely to find that their ventures are most effective when applied to their community rather than broadly to other geographic locales [58]. These observations highlight the regional characteristics of social entrepreneurship and its significance in comparison to entrepreneurship. However, to the best of the authors' knowledge, no study has quantitatively analyzed the differences and similarities between social entrepreneurship and entrepreneurship research from a regional perspective. Therefore, we referenced the methodology used in studies in the fields of "Steel structure" and "Greek construction project" [59,60] to examine the differences and similarities in international co-authorship networks.

### 2.4. Research Question Development

The current study conducted a comparative analysis of the social entrepreneurship and entrepreneurship research fields from the perspective of international co-authorship networks. The research questions are as follows:

- RQ1: What are the differences and similarities between the social entrepreneurship and entrepreneurship research fields based on the characteristics of international co-authorship networks?
- RQ2: Which international collaboration methodology is more prevalent between the entrepreneurship and social entrepreneurship research fields?
- RQ3: Which are the most collaborative and collaborating countries in the research field of social entrepreneurship and entrepreneurship?

### 3. Methods

#### 3.1. Bibliometric Analysis

We conducted a bibliometric analysis to analyze the basic information of articles, such as author, journal, keywords, and citations, to investigate the development process of the research field [12,27]. Since bibliometric analysis is based on a quantitative approach, it allows us to view the research field objectively, whereas conventional studies utilize literature reviews that mainly adopt a subjective perspective [61]. Bibliometric analysis includes a variety of approaches, such as bibliographic coupling analysis [12], co-citation analysis [6,12,30], and co-word analysis [30]. We conducted co-authorship analysis alongside bibliometric analysis while adopting the approach of social network analysis. Co-authorship is defined as interactions taken within a social context among multiple scientists that facilitate the sharing of meaning and accomplishing tasks with respect to a mutually shared goal [59]. The frequency and quality of communication allow us to identify the knowledge-sharing activities of different researchers when publishing a paper together [59]. Moreover, social network analysis identifies how problems are mitigated, how organizations are structured, how phenomena evolve, and how individuals and organizations prosper after achieving their aims [62]. The reason for adopting co-authorship analysis was to identify the strength of international collaboration through identifying authors' collaboration activity between countries.

#### 3.2. Data Collection

The flow of the analysis is shown in Figure 1. First, we extracted bibliographic data from social entrepreneurship and entrepreneurship research from the Web of Science database. The search keywords were TS = ("social entrepreneurship") and TS = ("entrepreneurship" NOT "social entrepreneurship"). This study examined articles published in English between 1999 and 2021. Finally, the search returned 1894 publications on social entrepreneurship and 27,616 publications on entrepreneurship. In total, 29,510 publications were extracted for data analysis.

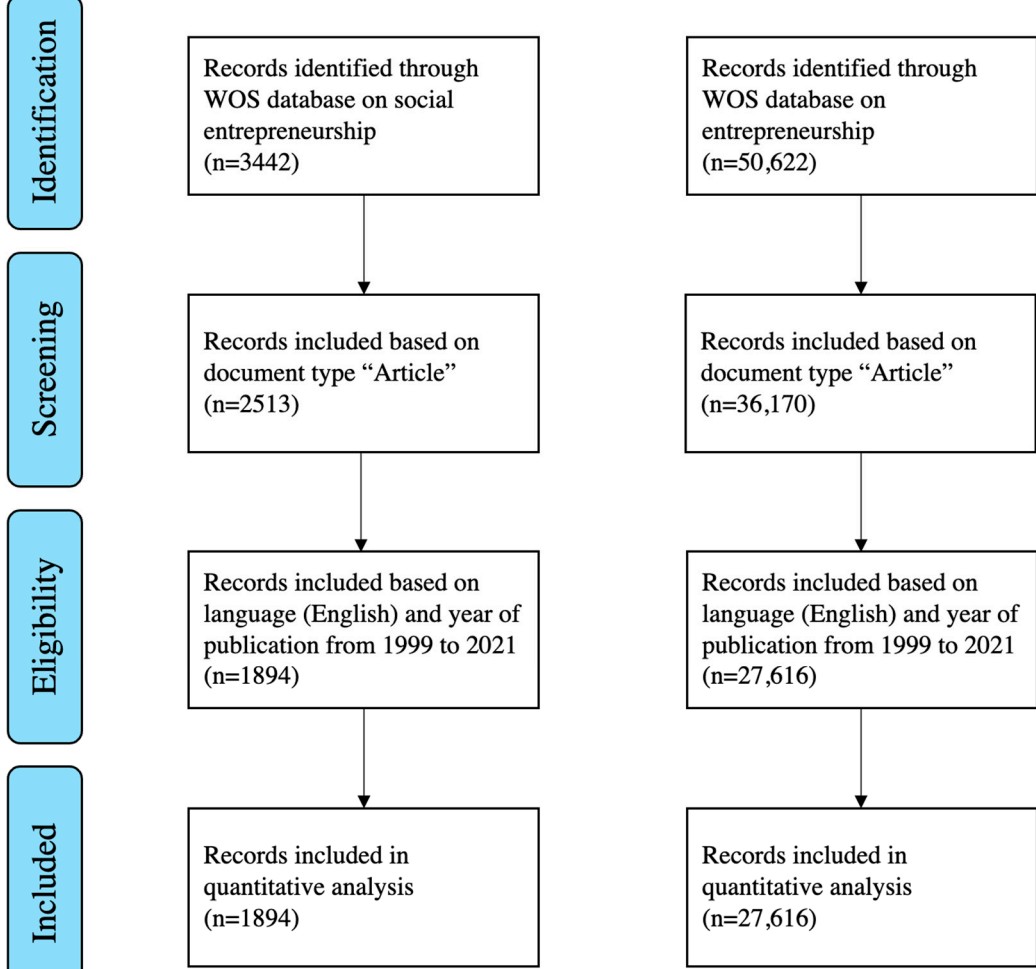

**Figure 1.** Data extraction process.

*3.3. Background Information*

3.3.1. Annual Trends of Publication

First, we visualized the annual trends of publications to identify the research field development process and understand the peaks in publications, which demonstrate the areas of interest of scholars in conducting research.

3.3.2. International Co-Authorship Network Development

Second, we visualized international co-authorship networks using bibliometric analysis. Bibliometrics analysis is typically utilized to represent how author, publication, and journal are related to one another [63]. Analyzing connectivity patterns between countries in international co-authorship networks allowed us to assess field-specific community structures [64]. In this study, we adopted VOSviewer (version 1.6.19). This is a bibliographic network visualization software developed by van Eck and Waltman. VOSviewer has been widely used for constructing and viewing bibliometric maps [63,65]. The size of a node represents the weight of an item [66]. The higher the weight, the larger the circle visualized in the network. For instance, the weight of the current study represents the number of papers published by institutions from more than two countries. The distance between nodes represents the relatedness of countries [66]. Relatedness demonstrates the frequency with which countries engage in international collaboration. As the setting for visualizing international co-authorship networks, we selected countries as the unit of analysis, full counting for the counting method, and deselected the function to ignore documents with a number of authors. We selected a default setting for the thresholds with the minimum

number of documents of an author set at 5 and the minimum number of citations of an author at 0. Finally, 65 countries met the thresholds for social entrepreneurship research, and 120 countries conducting entrepreneurship research were selected to visualize the international co-authorship network.

*3.4. Methodology*

3.4.1. Identifying the Characteristics of Co-Authorship Networks

To identify the characteristics of international co-authorship networks, we utilized three measures: degree, closeness, and betweenness centralization. For RQ1, we developed an adjacency matrix utilizing the bibliometrix package (version 4.1.3) in R (version 4.3.1). R is one of the most effective statistical software environments with an open source platform [67]. The bibliometrix package is a tool used to conduct quantitative research for bibliometric analysis [68]. We developed an adjacency matrix by simplifying the network with no weight and loop. Then, we defined the network as the maximum connected component, while excluding the isolated vertex to maintain consistency in the calculation of centralization. Finally, we calculated the degree centralization, closeness centralization, and betweenness centralization to identify network characteristics while utilizing the igraph package (version 1.5.1) in R to input the adjacency matrix for data analysis. The igraph package provides functions which allow us to build and analyze networks [69]. igraph is known to be specialized in conducting exploratory network analysis [70]. Centralization refers to a network measured by each node's deviation in centrality to indicate its tendency [71,72]. Notably, one influential node can impact the centralization value. Therefore, centralization can only represent the predisposition of a network. The degree centralization expresses the deviation of a node's degree centrality in a network [73]. Therefore, a higher degree centralization indicates that there are one or more nodes with a high degree centrality [73]. Closeness centralization identifies the tendencies of nodes with different degrees of closeness centrality [74]. Therefore, higher closeness centralization does not imply that all the nodes in the networks have high closeness centrality but rather represents the deviation of each node in the networks. Betweenness centralization identifies how a network is concentrated on each node with betweenness centrality [73]. Betweenness centralization was also considered as the indicator in analyzing the flow of information between networks through identifying the relationships between nodes [75].

3.4.2. Identification of Strong International Co-Authorship

For RQ2, to identify a stronger international co-authorship network between social entrepreneurship and entrepreneurship research, we utilized two measurements: average distance and density. The current study referred to a previous study which compared co-authorship networks between the individual, institutional, and national levels regarding "steel structure" [59]. We developed an adjacency matrix using the bibliometrix package and simplifying the network with no loop and weight. Then, we defined the network while extracting the maximum number of connected components and excluding isolated vertices. Finally, we calculated the average distance and density using the igraph package in R. The average distance indicates how close a randomly chosen country is to another country by steps. For example, if there is an average distance of 3, this implies that by reaching three other countries on average, one country can reach any other targeted country [59]. Therefore, when a network's average distance is shorter, it is more likely that countries in the network will be able to more easily collaborate with each other. Density indicates the actual connections among all potential connections between each node [71]. The current study identified the network with the maximum number of connected components while eliminating the isolated nodes in the network. A previous study utilized density to compare co-authorship networks regarding steel structures at the individual, institutional, and national levels [59]. Moreover, another study compared communication flow during group sessions to measure the degree of active discussion with and without a moderator in the group to foster communication between team members [59]. This study utilized

density to measure whether social entrepreneurship or entrepreneurship research has a strong international co-authorship network.

### 3.4.3. Identifying the Top Collaborative and Collaborating Countries

For RQ3, we identified the most collaborative and collaborating countries based on international co-authorship networks. Collaborative countries have a high number of connections in international co-authorship studies. Conversely, collaborating countries conduct a high number of international co-authorship studies. Comparing the most collaborative and collaborating countries in each research field allows for examining which countries are focused on conducting international co-authorship studies in terms of number of collaborations and number of connections between each country. The current study was based on a previous study that identified the most collaborative and collaborating countries [59]. In this study, we extracted the data through VOSviewer. To calculate the top collaborative and collaborating countries, we selected countries as the unit of analysis, full counting for the counting method, and deselected the function to ignore documents with a number of authors. We selected the minimum number threshold of documents produced by a country as 1 and the minimum number of citations of a country as 0. All in all, 103 countries met the thresholds for social entrepreneurship research, and 168 countries for entrepreneurship research. The current study followed the interpretation of a previous study that regarded the total link strength as the number of collaborations conducted by a country and the weight link as the number of connections that a country possesses. Moreover, a previous study identified the top collaborative countries in the international co-authorship network of the "steel structure" research field [59]. Identifying collaborative and collaborating countries allowed us to identify the countries that are leading in this research field.

## 4. Results

### 4.1. Background Information
Annual Trends of Publication

The annual trends of the publications for social entrepreneurship and entrepreneurship research are presented in Figure 2. The starting year for this study was 1999 based on when the first article on the topic of social entrepreneurship extracted from the Web of Science database was published [76]. We visualized the annual publication trends based on the data extracted from a library called bibliometrix in R. The procedure was as follows. First, we inserted the bibliographic information into R. Second, the bibliographic file was converted and imported using the convert2df function. Lastly, we conducted a descriptive analysis with the biblioAnalysis function to calculate and utilize the summary function to summarize the results. Social entrepreneurship research first received scholarly interest in 2016. Entrepreneurship research has also gradually increased their number of publications over the 19 years since 1999. From 2018 onwards, the number of publications rapidly increased.

### 4.2. International Co-Authorship Network Development
4.2.1. Social Entrepreneurship

Figure 3 presents the international co-authorship network for social entrepreneurship, which consists of 65 countries in nine clusters. Table A1 represents country which belong to each clusters are listed with the number of connection that each country own in the network is shown. All the clusters contain countries from different continents, cultures, and languages, implying that the relatedness of the research topic encouraged researchers to conduct international co-authorship. For instance, cluster 2 (green) consists of 12 countries in Oceania, East Asia, Europe, South Asia, Southeast Asia, and Southeastern Europe, demonstrating the connection between developed countries and developing countries in conducting international collaboration studies on social entrepreneurship. From a subjective perspective, the co-authorship network for social entrepreneurship research is

widely distributed, and the distance between countries is relatively longer compared to the international co-authorship network for entrepreneurship.

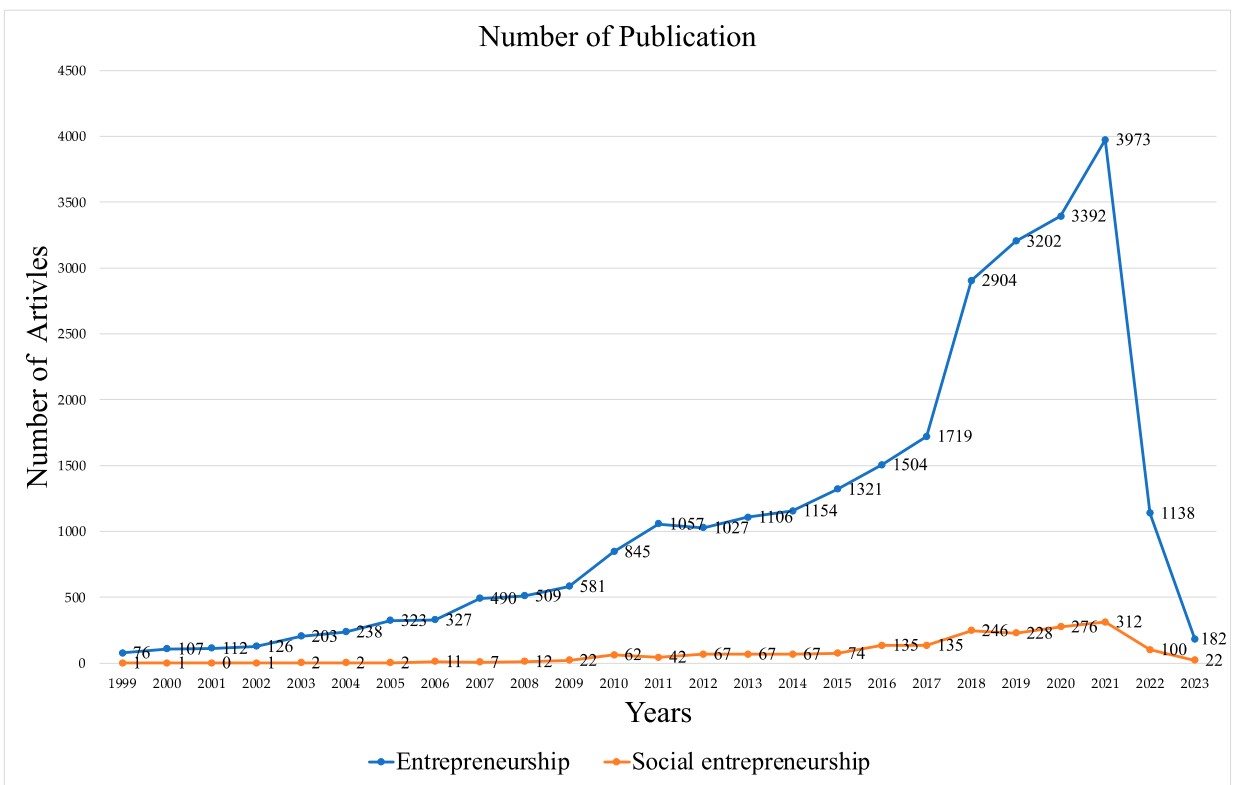

**Figure 2.** Trends in number of publications.

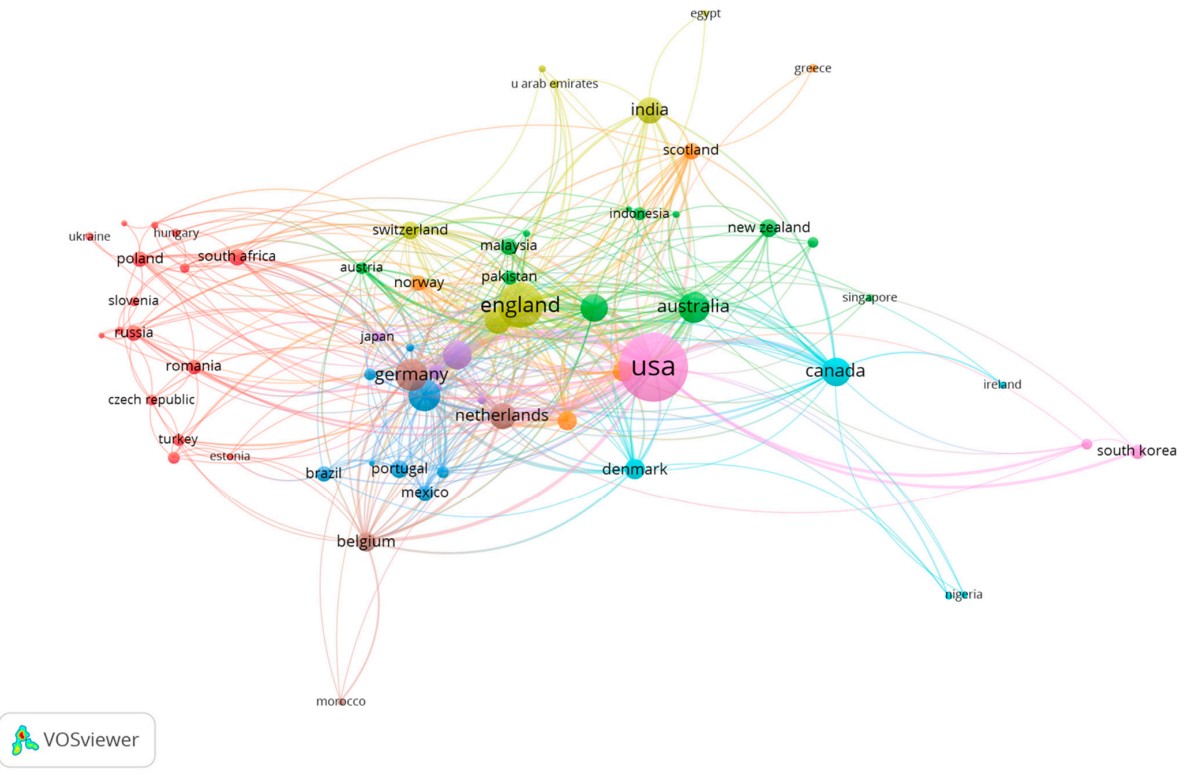

**Figure 3.** International co-authorship network of social entrepreneurship (1999–2021).

4.2.2. Entrepreneurship

Figure 4 presents the international co-authorship network for entrepreneurship. Table A2 represents country which belong to each clusters are listed with the number of connection that each country own in the network is shown. The network consists of 120 countries in 10 clusters. The characteristics of connection between developed and developing countries collaborating internationally are distinctively visualized. Few clusters contain countries that are geographically closer to one other. For instance, consider cluster 3 (blue) from Africa, cluster 4 (yellow) from South America, cluster 5 (purple), and cluster 6 (light blue) from Europe. This unique collaboration could be owing to the relatedness of the research topics they share in conducting international co-authorship studies. From a subjective perspective, the co-authorship network in the entrepreneurship research field is centralized compared with that in the social entrepreneurship research field. Additionally, the distance between nodes seems to be shorter compared to social entrepreneurship's international co-authorship network, implying that entrepreneurship involves stronger international collaboration.

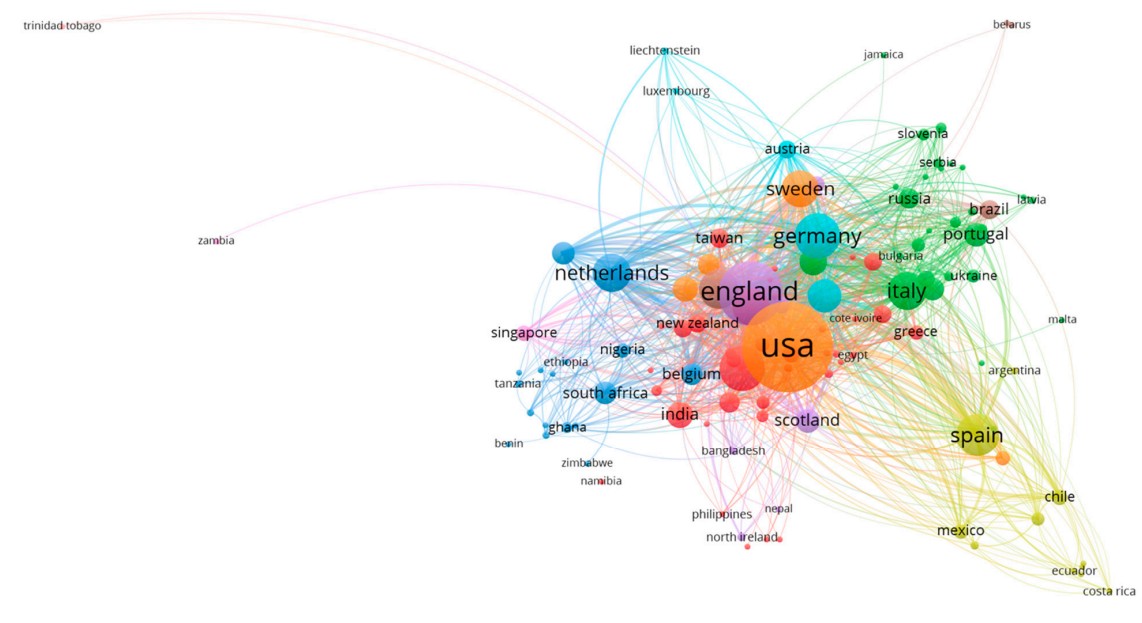

**Figure 4.** International co-authorship network of entrepreneurship (1999–2021).

*4.3. International Co-Authorship Network Analysis*

Table 3 presents the results for RQ1. We identified three categories of centralization, including degree, closeness, and betweenness centralization. The degree of degree centralization is greater in entrepreneurship research. This implies that more than one country has a higher degree of connection with other countries in conducting co-authorship studies compared with the social entrepreneurship research field. Moreover, entrepreneurship research fields have higher degree of closeness centralization. This implies that there is more than one node in the entrepreneurship research field that has a closer connection to multiple countries compared to the social entrepreneurship research field. This suggests that it is easier to collaborate internationally in the entrepreneurship research field. Finally, betweenness centralization represents how one or more countries mediate subgroups in the international co-authorship network. Betweenness centralization is greater in the social entrepreneurship research field than the entrepreneurship research field, which suggests that there is more than one country in the social entrepreneurship research field that is an active mediator of subgroups compared to the entrepreneurship research field.

**Table 3.** International co-authorship network structure for social entrepreneurship and entrepreneurship.

| Measures | Social Entrepreneurship | Entrepreneurship |
|---|---|---|
| Degree centralization | 0.454467 | 0.597905 |
| Closeness centralization | 0.474295 | 0.587200 |
| Betweenness centralization | 0.210264 | 0.145042 |

### 4.4. Identification of Strong International Co-Authorship

To quantitatively analyze strong international co-authorship in the social entrepreneurship and entrepreneurship networks for RQ2, the average distance and density were measured to determine the relationship between the nodes. Average distance indicates the speed and flow of information from sender to receiver. As shown in Table 4, entrepreneurship has a shorter average distance than social entrepreneurship. This indicates that international co-authorship networks for entrepreneurship research require fewer than two countries to reach any other country in the network on average. However, the social entrepreneurship research field requires more than two countries to mediate these relationships before reaching the targeted country. Density is greater in the entrepreneurship research field, which indicates that there is a higher degree of connectivity in entrepreneurship networks than in social entrepreneurship networks. Moreover, a higher density implies frequent information sharing and communication among countries. Therefore, this study found that the entrepreneurship research field conducts more international co-authorship studies.

**Table 4.** Strong international co-authorship for social entrepreneurship and entrepreneurship.

| Measures | Social Entrepreneurship | Entrepreneurship |
|---|---|---|
| Average Distance | 2.244845 | 1.975059 |
| Density | 0.108033 | 0.165609 |

### 4.5. Identifying the Top Collaborative Countries

Table 5 presents the top collaborative countries, which include countries with the highest number of connections with other countries in conducting international co-authorship, to answer RQ3. Appendix B represents a network with the lists of country which belong to each clusters and the number of connection that each country own in the network to calculate the collaborative country for RQ3 is presented. In The most collaborative country in the entrepreneurship research field was England, which had connections with 120 countries, followed by the USA and Germany, with 115 and 88 countries, respectively. The most collaborative countries in social entrepreneurship research were the USA, which had connections with 55 countries in conducting international co-authorship, and England and Germany, with 54 and 39 countries, respectively.

### 4.6. Identifying the Top Collaborating Countries

Table 6 presents the top collaborating countries and describes how many times each country conducted an international co-authorship study to answer RQ3. Appendix B represents a network with the lists of country which belong to each clusters and the number of connection that each country own in the network to calculate the collaborating country for RQ3 is presented. The international co-authorship network which was developed through conducting RQ3 is represented in the Appendix B. For entrepreneurship, the country with the highest number of international co-authorship studies conducted was the USA (4547 times), followed by England and Germany (3822 times and 1892 times, respectively). Entrepreneurship research was mainly led by countries from North America, Northwest Europe, Europe, Oceania, Western Europe, East Asia, Southern Europe, Southwestern Europe, Northern Europe, and South Asia. Countries from South America were not included in the top 20 collaborating countries in the entrepreneurship research field. For

social entrepreneurship, the country that conducted the highest number of international co-authorship studies was the USA at 309 times, followed by England and Germany at 253 and 128 times, respectively. There were a number of developed countries among the top 20 collaborating countries. This implies that the study of social entrepreneurship has been mainly led by developed countries.

**Table 5.** Top collaborative countries: country name and number of collaborators (left: social entrepreneurship, right: entrepreneurship).

| | Social Entrepreneurship | | | Entrepreneurship | |
|---|---|---|---|---|---|
| | Country | Number of Collaborators | | Country | Number of Collaborators |
| 1 | USA | 55 | 1 | England | 120 |
| 2 | England | 54 | 2 | USA | 115 |
| 3 | Germany | 39 | 3 | Germany | 88 |
| 4 | Netherlands | 37 | 4 | Canada | 86 |
| 5 | France | 35 | 5 | Italy | 84 |
| 6 | Spain | 33 | 6 | France | 82 |
| 7 | Australia | 32 | 7 | Spain | 80 |
| 8 | China | 30 | 8 | China | 78 |
| 9 | Canada | 27 | 9 | Australia | 76 |
| 10 | Belgium | 26 | 10 | Netherlands | 75 |
| 10 | Finland | 26 | 11 | Finland | 70 |
| 10 | Switzerland | 26 | 12 | Sweden | 65 |
| 13 | Austria | 24 | 13 | Poland | 63 |
| 14 | Italy | 23 | 14 | Denmark | 62 |
| 15 | Scotland | 21 | 15 | Scotland | 61 |
| 16 | Denmark | 20 | 16 | India | 60 |
| 16 | Sweden | 20 | 16 | Portugal | 60 |
| 18 | Norway | 19 | 18 | Belgium | 59 |
| 18 | Poland | 19 | 18 | Switzerland | 59 |
| 20 | Mexico | 18 | 20 | South Africa | 58 |

**Table 6.** Top collaborating countries: country name and number of collaborations (left: social entrepreneurship, right: entrepreneurship).

| | Social Entrepreneurship | | | Entrepreneurship | |
|---|---|---|---|---|---|
| | Country | Number of Collaborators | | Country | Number of Collaborators |
| 1 | USA | 309 | 1 | USA | 4547 |
| 2 | England | 253 | 2 | England | 3822 |
| 3 | Germany | 128 | 3 | Germany | 1892 |
| 4 | France | 116 | 4 | Netherlands | 1469 |
| 5 | Netherlands | 105 | 5 | France | 1403 |
| 6 | Australia | 93 | 6 | Canada | 1335 |
| 6 | China | 93 | 7 | China | 1334 |
| 8 | Spain | 89 | 8 | Italy | 1262 |
| 9 | Canada | 80 | 9 | Spain | 1240 |
| 10 | Italy | 77 | 10 | Sweden | 1207 |
| 11 | Finland | 74 | 11 | Australia | 1194 |
| 12 | Belgium | 73 | 12 | Denmark | 762 |
| 13 | Denmark | 56 | 13 | Finland | 741 |
| 14 | Sweden | 54 | 14 | Belgium | 703 |
| 15 | Switzerland | 52 | 15 | Switzerland | 692 |
| 16 | Austria | 51 | 16 | Scotland | 588 |
| 17 | Scotland | 47 | 17 | Austria | 486 |
| 18 | India | 43 | 18 | Norway | 438 |
| 19 | Colombia | 36 | 19 | Portugal | 397 |
| 19 | Norway | 36 | 20 | India | 348 |

## 5. Discussion

### 5.1. Theoretical Implications

Considering different degrees of international co-authorship in entrepreneurship and social entrepreneurship research, a previous study found that social entrepreneurship was focused on regional economic growth during the global financial crisis in Korea in the early 2000s [77]. Similarly, social entrepreneurship for the Regional Comprehensive Economic Partnership (RCEP) in Southeast Asia was found to be an effective means of promoting sustainable development based on empirical analysis [78]. These findings suggest that social entrepreneurship is suitable for developing regional economies. On the other hand, a previous study on entrepreneurship as an independent variable [79] and a moderating variable [80] found that it has a positive and significant influence on regional economic development. However, entrepreneurship has been linked to globalization as a prerequisite for entrepreneurial success [81]. According to Prashantham et al. (2018), globalization facilitates technology entrepreneurship through fostering opportunity with innovation [81]. From the perspective of international entrepreneurship, it is known that when a firm effectively responds to the challenges of internationalization, it positively influences performance [82]. The discussion above indicates that both social entrepreneurship and entrepreneurship contribute to mitigating regional challenges and developing economic value. However, globalization is known to contribute to effective monetization and large-scale economy growth for entrepreneurship [83]. On the other hand, acquiring knowledge regarding local community and societal problems is significant in social entrepreneurship for value creation [49]. Therefore, following the findings for RQ1 and RQ2, entrepreneurship and globalization are inseparable, and social entrepreneurship has the most access to localization.

### 5.2. Practical Implications

Previous studies which utilized bibliometric analysis mainly focused on analyzing social entrepreneurship and entrepreneurship separately. A previous study found that microcredit organizations were able to provide opportunities for people who were suffering from poverty in a developing country [9]. Moreover, a previous study stated that one characteristic of social entrepreneurship is the aim to seek embeddedness and create close relationships with one's community [58]. This study advances the understanding of the boundary between social entrepreneurship and entrepreneurship research in three new ways. For RQ1, while conducting bibliometric analysis simultaneously for comparison, we identified the differences and similarities between the entrepreneurship and social entrepreneurship research fields. Results for degree, closeness, and betweenness centralization revealed that entrepreneurship research has more nodes that possess a higher number of connections and fosters a closer relationship with other countries compared to the social entrepreneurship research field. Regarding betweenness centralization, social entrepreneurship research has a higher degree of connections, indicating a higher tendency to be a mediator of subgroups within the network. Second, for RQ2, we found that entrepreneurship research leads to stronger international collaboration efforts compared to social entrepreneurship, following a previous study which stated that globalization allows entrepreneurship to be successful [81]. After calculating average distance and density, the findings indicated that entrepreneurship international co-authorship networks have a shorter average distance and higher density. Therefore, the current study revealed that the entrepreneurship research field has stronger international co-authorship efforts than social entrepreneurship. Third, to examine RQ3, this study determined the most collaborative and collaborating countries in the social entrepreneurship and entrepreneurship research fields. The results indicate that both research fields are primarily led by developed nations. This is owing to social entrepreneurship's focus on regional challenges, which limits international collaboration for disseminating research globally. This discussion validates the findings of previous studies, which revealed that social entrepreneurship is effective in addressing regional challenges. In practical terms, when initiating a project within the social

entrepreneurship research field, it is important to focus on local and regional challenges. On the other hand, when conducting programs within the field of entrepreneurship, it is important to invite entrepreneurs who have the experience in collaborating with other entrepreneurs and institutions from different countries to share their experience of interacting with people from different countries to solve challenges together while gaining a new perspective. Lastly, based on the results for RQ1, it was revealed that social entrepreneurship research has a higher degree of betweenness centralization, implying that regional challenges in social entrepreneurship are more difficult to transfer from one country to another compared to technology and knowledge transfer in the entrepreneurship research field. This is because they have complex and different historical backgrounds which are deeply connected with regional challenges. However, collaborating internationally while focusing on regional challenges can mitigate such challenges in innovative ways though attracting the attention of investors from different countries and disseminating solutions globally. The analysis of RQ2 revealed that social entrepreneurship research has relatively weaker international co-authorship compared to entrepreneurship research. This indicates that social entrepreneurship research and activities are mainly conducted by people in communities, regions, and countries. Therefore, to effectively mitigate the challenges of social entrepreneurship, it is important to engage with the community and people who are affected by the challenges in each region. According to the analysis conducted for RQ3, developed countries had the highest number of international co-authorships in both the social entrepreneurship and entrepreneurship research field. Therefore, to foster social entrepreneurship, it is important to foster international co-authorship among researchers and practitioners in developing countries to promote active research.

## 6. Conclusions

This study aimed at identifying the boundaries between social entrepreneurship and entrepreneurship through a comparative analysis of international co-authorship networks. We extracted bibliographic information from the Web of Science and conducted a bibliometric analysis using VOSviewer and the R programming language. The number of publications in the fields of social entrepreneurship and entrepreneurship dramatically increased after 2018. The visualization of international co-authorship networks implied that entrepreneurship research has centralized networks, while social entrepreneurship research has decentralized networks. We further analyzed the degree centralization, betweenness centralization, and closeness centralization using the R programming language. Consistent with previous findings, the current study found that social entrepreneurship is rooted in both regional and local challenges. Moreover, this study is the first to quantitatively confirm that social entrepreneurship research engages less in international collaboration compared to entrepreneurship research. Furthermore, the analysis conducted for RQ3 revealed that international co-authorship in social entrepreneurship and entrepreneurship research is primarily conducted by developed countries. This implies that despite its focus on local challenges, social entrepreneurship involves collaboration with developed and developing countries to combat social issues. Based on the results, in social entrepreneurship education, it is more appropriate to focus on topics related to the local community, to which students feel more attached. However, in entrepreneurship education, it is appropriate to choose case studies and guest speakers who are successful in the global field and explain how ideas can be developed in a collaborative effort by communicating with people from different backgrounds and cultures to develop and foster innovation. This study has four limitations. First, there is a disparity in the number of articles between the research fields of social entrepreneurship and entrepreneurship, with entrepreneurship research being more extensive. Second, the current study only used the Web of Science database, which limits the inclusion of relevant publications from other databases, such as Scopus. Moreover, only articles published in English were considered, excluding publications in other languages [84], due to the lack of the authors' ability to understand articles other than those written in English in detail. Third, the current study conducted a bibliometric analysis with data extracted

from different networks. To Lastly, there may be projects in the social entrepreneurship and entrepreneurship research fields that were not included in our analysis as they did not become academic articles. Furthermore, the research objective established for the current study was focused on quantitatively identifying the differences and similarities between social entrepreneurship and entrepreneurship from regional aspects via an international co-authorship analysis. The current study was able to identify that the regional aspects of social entrepreneurship and entrepreneurship are in line with the results of previous studies [77,78,81,83]. However, future analyses should broaden the data extraction methodology and incorporate case studies to analyze international collaborations within social entrepreneurship and entrepreneurship phenomena.

**Author Contributions:** Conceptualization, K.K. (Karin Kurata), Y.J.L., K.K. (Kota Kodama), and S.S; methodology, S.M. and K.K. (Kota Kodama); software, K.K. (Karin Kurata) and S.M.; validation, S.M., S.S., and K.K. (Kota Kodama); formal analysis, Y.J.L. and K.K. (Kota Kodama); data curation, K.K. (Karin Kurata) and S.M.; writing—original draft preparation, K.K. (Karin Kurata); writing—review and editing, Y.J.L. and K.K. (Kota Kodama); visualization, K.K. (Karin Kurata) and S.M.; supervision, Y.J.L. and K.K (Kota Kodama); project administration, Y.J.L. All authors have read and agreed to the published version of the manuscript.

**Funding:** This research was funded by [Grants-in-Aid for Scientific Research] grant number [20H01546] and [Grant-in-Aid for Early-Career Scientists] grant number [22K13466]. The funding sources did not participate in the study design, data collection, analysis, interpretation, report writing, or the decision to submit this article for publication.

**Institutional Review Board Statement:** Not applicable.

**Informed Consent Statement:** Not applicable.

**Data Availability Statement:** Data sharing not applicable.

**Conflicts of Interest:** The authors declare no conflict of interest.

## Appendix A. Network Data Extracted for International Co-Authorship Network Development

**Table A1.** Top countries by cluster for number of connections to form international co-authorship (social entrepreneurship).

| Cluster 1 (Red) | | | Cluster 2 (Green) | | | Cluster 3 (Blue) | | |
|---|---|---|---|---|---|---|---|---|
| Rank | Country | Weight | Rank | Country | Weight | Rank | Country | Weight |
| 1 | Poland | 19 | 1 | Australia | 30 | 1 | Spain | 29 |
| 2 | Russia | 14 | 2 | China | 27 | 2 | Colombia | 16 |
| 3 | Croatia | 12 | 3 | Austria | 21 | 3 | Chile | 15 |
| 4 | South Africa | 11 | 4 | New Zealand | 12 | 3 | Mexico | 15 |
| 5 | Czech Republic | 10 | 4 | Pakistan | 12 | 5 | Saudi Arabia | 12 |
| 6 | Lithuania | 9 | 6 | Malaysia | 10 | 6 | Portugal | 11 |
| 6 | Turkey | 9 | 7 | North Macedonia | 9 | 7 | Brazil | 10 |
| 8 | Estonia | 8 | 8 | Wales | 7 | 8 | Ecuador | 8 |
| 9 | Romania | 7 | 9 | Philippines | 6 | | | |
| 10 | Iran | 6 | 10 | Vietnam | 5 | | | |
| 10 | Slovakia | 6 | 11 | Indonesia | 4 | | | |
| 12 | Hungary | 5 | 12 | Singapore | 3 | | | |
| 13 | Kazakhstan | 4 | | | | | | |
| 13 | Slovenia | 4 | | | | | | |
| 15 | Latvia | 2 | | | | | | |
| 15 | Ukraine | 2 | | | | | | |

**Table A1.** *Cont.*

| | Cluster 4 (Yellow) | | | Cluster 5 (Purple) | | | Cluster 6 (Light Blue) | |
|---|---|---|---|---|---|---|---|---|
| Rank | Country | Weight | Rank | Country | Weight | Rank | Country | Weight |
| 1 | England | 47 | 1 | Italy | 23 | 1 | Canada | 24 |
| 2 | France | 30 | 2 | Taiwan | 12 | 2 | Denmark | 19 |
| 3 | Switzerland | 22 | 3 | Bangladesh | 9 | 3 | Nigeria | 4 |
| 4 | India | 14 | 3 | Japan | 9 | 3 | Uganda | 4 |
| 5 | UAE | 7 | 5 | Thailand | 5 | 5 | Ireland | 2 |
| 6 | Lebanon | 4 | | | | | | |
| 7 | Egypt | 2 | | | | | | |
| | Cluster 7 (Orange) | | | Cluster 8 (Brown) | | | Cluster 9 (Pink) | |
| Rank | Country | Weight | Rank | Country | Weight | Rank | Country | Weight |
| 1 | Finland | 22 | 1 | Germany | 37 | 1 | USA | 46 |
| 2 | Scotland | 20 | 2 | Netherlands | 29 | 2 | Israel | 6 |
| 3 | Sweden | 19 | 3 | Belgium | 24 | 3 | South Korea | 3 |
| 4 | Norway | 17 | 4 | Morocco | 4 | | | |
| 5 | Greece | 2 | | | | | | |

**Table A2.** Top countries by cluster for number of connections to form international co-authorship (entrepreneurship).

| | Cluster 1 (Red) | | | Cluster 2 (Green) | | | Cluster 3 (Blue) | |
|---|---|---|---|---|---|---|---|---|
| Rank | Country | Weight | Rank | Country | Weight | Rank | Country | Weight |
| 1 | Australia | 76 | 1 | Italy | 80 | 1 | Netherlands | 73 |
| 2 | China | 74 | 2 | Finland | 68 | 2 | Belgium | 59 |
| 3 | India | 58 | 3 | Poland | 63 | 3 | Switzerland | 58 |
| 4 | Malaysia | 55 | 4 | Portugal | 58 | 4 | South Africa | 56 |
| 4 | Turkey | 55 | 5 | Russia | 55 | 5 | Ghana | 35 |
| 6 | New Zealand | 51 | 6 | Czech Republic | 43 | 6 | Nigeria | 34 |
| 6 | UAE | 51 | 7 | Hungary | 41 | 7 | Kenya | 23 |
| 8 | Pakistan | 46 | 8 | Romania | 38 | 7 | Tanzania | 23 |
| 9 | Saudi Arabia | 43 | 9 | Lithuania | 35 | 9 | Uganda | 20 |
| 10 | Greece | 41 | 10 | Croatia | 33 | 10 | Ethiopia | 16 |
| 10 | Iran | 41 | 11 | Slovenia | 32 | 11 | Cameroon | 12 |
| 10 | Japan | 41 | 12 | Slovakia | 28 | 12 | Malawi | 11 |
| 13 | South Korea | 38 | 13 | Estonia | 27 | 13 | Rwanda | 10 |
| 14 | Taiwan | 37 | 13 | Ukraine | 27 | 14 | Botswana | 8 |
| 15 | Egypt | 35 | 15 | Serbia | 26 | 15 | Zimbabwe | 7 |
| 16 | Indonesia | 34 | 16 | Bosnia and Herzegovina | 23 | 16 | Democratic Republic of Congo | 6 |
| 16 | Thailand | 34 | 17 | Iceland | 22 | 17 | Benin | 5 |
| 18 | Vietnam | 32 | 17 | Morocco | 22 | | | |
| 19 | Oman | 30 | 19 | Latvia | 18 | | | |
| 20 | Cyprus | 28 | 20 | Albania | 17 | | | |
| 21 | North Macedonia | 25 | 20 | Bulgaria | 17 | | | |
| 22 | Qatar | 24 | 22 | Kosovo | 12 | | | |
| 22 | Tunisia | 24 | 23 | Jamaica | 9 | | | |
| 24 | Kazakhstan | 23 | 23 | Macedonia | 9 | | | |
| 25 | Philippines | 22 | 23 | Malta | 9 | | | |
| 26 | Lebanon | 16 | 26 | Azerbaijan | 8 | | | |
| 27 | Bahrain | 15 | 27 | Georgia | 6 | | | |
| 27 | Jordan | 15 | 27 | Montenegro | 6 | | | |
| 29 | Kuwait | 12 | | | | | | |
| 29 | Sri Lanka | 12 | | | | | | |
| 31 | Iraq | 11 | | | | | | |
| 32 | Brunei | 10 | | | | | | |

**Table A2.** *Cont.*

| Cluster 1 (Red) | | | Cluster 2 (Green) | | | Cluster 3 (Blue) | | |
| --- | --- | --- | --- | --- | --- | --- | --- | --- |
| Rank | Country | Weight | Rank | Country | Weight | Rank | Country | Weight |
| 33 | Kyrgyzstan | 8 | | | | | | |
| 34 | Afghanistan | 7 | | | | | | |
| 34 | Fiji | 7 | | | | | | |
| 36 | Cote Ivoire | 6 | | | | | | |
| 36 | Palestine | 6 | | | | | | |
| 36 | Senegal | 6 | | | | | | |
| 36 | Yemen | 6 | | | | | | |
| 40 | Namibia | 3 | | | | | | |

| Cluster 4 (Yellow) | | | Cluster 5 (Purple) | | | Cluster 6 (Light Blue) | | |
| --- | --- | --- | --- | --- | --- | --- | --- | --- |
| Rank | Country | Weight | Rank | Country | Weight | Rank | Country | Weight |
| 1 | Spain | 74 | 1 | England | 106 | 1 | Germany | 85 |
| 2 | Chile | 40 | 2 | Scotland | 60 | 2 | France | 76 |
| 3 | Mexico | 39 | 3 | Ireland | 53 | 3 | Austria | 56 |
| 4 | Colombia | 33 | 4 | Wales | 47 | 4 | Liechtenstein | 17 |
| 4 | Peru | 33 | 5 | Bangladesh | 33 | 4 | Luxembourg | 17 |
| 6 | Argentina | 22 | 6 | Nepal | 13 | | | |
| 7 | Ecuador | 17 | 6 | North Ireland | 13 | | | |
| 8 | Costa Rica | 12 | | | | | | |
| 9 | Venezuela | 8 | | | | | | |
| 10 | Cuba | 7 | | | | | | |
| 10 | Uruguay | 7 | | | | | | |

| Cluster 7 (Orange) | | | Cluster 8 (Brown) | | | Cluster 9 (Color) | | |
| --- | --- | --- | --- | --- | --- | --- | --- | --- |
| Rank | Country | Weight | Rank | Country | Weight | Rank | Country | Weight |
| 1 | USA | 101 | 1 | Canada | 84 | 1 | Singapore | 43 |
| 2 | Sweden | 64 | 2 | Brazil | 39 | 2 | Zambia | 6 |
| 3 | Denmark | 62 | 3 | Cambodia | 8 | | | |
| 4 | Norway | 44 | 4 | Belarus | 7 | | | |
| 5 | Israel | 26 | | | | | | |

| Cluster 10 (color) | | |
| --- | --- | --- |
| Rank | Country | Weight |
| 1 | Trinidad and Tobago | 6 |

## Appendix B. Network Data Extracted in Analyzing the Top Collaborative and Top Collaborating Countries for RQ3

**Table A3.** Top countries by cluster for number of connections to form international co-authorship (social entrepreneurship).

| Cluster 1 | | | Cluster 2 | | | Cluster 3 | | |
| --- | --- | --- | --- | --- | --- | --- | --- | --- |
| Rank | Country | Weight | Rank | Country | Weight | Rank | Country | Weight |
| 1 | Austria | 24 | 1 | Russia | 14 | 1 | China | 30 |
| 2 | Poland | 19 | 2 | Portugal | 11 | 2 | Switzerland | 26 |
| 3 | Croatia | 16 | 3 | Brazil | 10 | 3 | Saudi Arabia | 14 |
| 4 | New Zealand | 14 | 3 | Czech Republic | 10 | 4 | Pakistan | 13 |
| 5 | Lithuania | 11 | 5 | Turkey | 9 | 5 | Taiwan | 12 |
| 6 | Hungary | 8 | 6 | Estonia | 8 | 6 | Malaysia | 11 |
| 7 | Serbia | 6 | 7 | Romania | 7 | 7 | UAE | 8 |
| 8 | Albania | 5 | 8 | Iran | 6 | 8 | Lebanon | 6 |
| 9 | Ethiopia | 3 | 8 | Slovakia | 6 | 9 | Jordan | 4 |

**Table A3.** *Cont.*

| | **Cluster 1** | | | **Cluster 2** | | | **Cluster 3** | |
|---|---|---|---|---|---|---|---|---|
| **Rank** | **Country** | **Weight** | **Rank** | **Country** | **Weight** | **Rank** | **Country** | **Weight** |
| 10 | Latvia | 2 | 10 | Slovenia | 5 | 10 | Uzbekistan | 3 |
| 10 | Monaco | 2 | 11 | Kazakhstan | 4 | 11 | Iraq | 2 |
| 10 | Ukraine | 2 | 12 | Montenegro | 1 | 12 | Brunei | 1 |
| 13 | Bosnia and Herzegovina | 1 | | | | | | |

| | **Cluster 4** | | | **Cluster 5** | | | **Cluster 6** | |
|---|---|---|---|---|---|---|---|---|
| **Rank** | **Country** | **Weight** | **Rank** | **Country** | **Weight** | **Rank** | **Country** | **Weight** |
| 1 | Netherlands | 37 | 1 | Spain | 33 | 1 | Italy | 23 |
| 2 | Canada | 27 | 2 | Mexico | 18 | 2 | Sweden | 20 |
| 3 | Norway | 19 | 3 | Colombia | 17 | 3 | Bangladesh | 10 |
| 4 | Tanzania | 8 | 3 | South Africa | 13 | 3 | Japan | 10 |
| 5 | Nigeria | 6 | 5 | Ecuador | 9 | 5 | Oman | 5 |
| 5 | Uganda | 6 | 5 | Iceland | 9 | 5 | Thailand | 5 |
| 7 | Kenya | 5 | 7 | Bolivia | 2 | 7 | Ghana | 4 |
| 8 | Malawi | 4 | 7 | Nicaragua | 2 | | | |
| 8 | Qatar | 4 | 7 | Zimbabwe | 2 | | | |
| 10 | Ireland | 3 | 10 | Botswana | 1 | | | |
| 11 | Cameroon | 1 | | | | | | |

| | **Cluster 7** | | | **Cluster 8** | | | **Cluster 9** | |
|---|---|---|---|---|---|---|---|---|
| **Rank** | **Country** | **Weight** | **Rank** | **Country** | **Weight** | **Rank** | **Country** | **Weight** |
| 1 | USA | 55 | 1 | England | 54 | 1 | India | 16 |
| 2 | Israel | 6 | 2 | Chile | 16 | 2 | Bahrain | 4 |
| 3 | South Korea | 3 | 3 | Cuba | 2 | 2 | Tunisia | 4 |
| 4 | Benin | 1 | 4 | Gambia | 1 | 3 | Egypt | 3 |
| 4 | Costa Rica | 1 | 4 | North Ireland | 1 | 5 | Cyprus | 2 |
| 4 | Kuwait | 1 | | | | | | |

| | **Cluster 10** | | | **Cluster 11** | | | **Cluster 12** | |
|---|---|---|---|---|---|---|---|---|
| **Rank** | **Country** | **Weight** | **Rank** | **Country** | **Weight** | **Rank** | **Country** | **Weight** |
| 1 | Germany | 39 | 1 | Australia | 32 | 1 | North Macedonia | 9 |
| 2 | France | 35 | 2 | Vietnam | 5 | 2 | Wales | 7 |
| 3 | Finland | 26 | 3 | Singapore | 3 | 3 | Philippines | 6 |
| 4 | Liechtenstein | 6 | 4 | Cambodia | 2 | 4 | Indonesia | 4 |
| 5 | Armenia | 3 | | | | | | |

| | **Cluster 13** | | | **Cluster 14** | | | **Cluster 15** | |
|---|---|---|---|---|---|---|---|---|
| **Rank** | **Country** | **Weight** | **Rank** | **Country** | **Weight** | **Rank** | **Country** | **Weight** |
| 1 | Belgium | 26 | 1 | Denmark | 20 | 1 | Scotland | 21 |
| 2 | Morocco | 5 | 2 | Zambia | 2 | 2 | Greece | 2 |
| 3 | Peru | 2 | | | | | | |

| | **Cluster 16** | | | **Cluster 17** | |
|---|---|---|---|---|---|
| **Rank** | **Country** | **Weight** | **Rank** | **Country** | **Weight** |
| 1 | Azerbaijan | 0 | 1 | Bulgaria | 0 |

**Table A4.** Top countries by cluster for number of connections to form international co-authorship (entrepreneurship).

| Cluster 1 | | | Cluster 2 | | | Cluster 3 | | |
|---|---|---|---|---|---|---|---|---|
| Rank | Country | Weight | Rank | Country | Weight | Rank | Country | Weight |
| 1 | China | 78 | 1 | Switzerland | 59 | 1 | Poland | 63 |
| 2 | Australia | 76 | 2 | South Africa | 58 | 2 | Russia | 56 |
| 3 | Malaysia | 56 | 3 | Hungary | 42 | 3 | Turkey | 55 |
| 4 | UAE | 52 | 4 | Ghana | 38 | 4 | Czech Republic | 43 |
| 5 | Pakistan | 46 | 5 | Nigeria | 35 | 5 | Romania | 39 |
| 6 | Saudi Arabia | 44 | 6 | Tanzania | 24 | 6 | Lithuania | 35 |
| 7 | South Korea | 40 | 7 | Kenya | 23 | 7 | Croatia | 33 |
| 7 | Taiwan | 40 | 7 | Morocco | 23 | 8 | Slovenia | 32 |
| 9 | Indonesia | 34 | 7 | Uganda | 23 | 9 | Slovakia | 28 |
| 9 | Thailand | 34 | 10 | Ethiopia | 18 | 9 | Ukraine | 28 |
| 11 | Vietnam | 32 | 11 | Cameroon | 13 | 11 | Estonia | 27 |
| 12 | North Macedonia | 25 | 12 | Malawi | 12 | 12 | Serbia | 26 |
| 13 | Kazakhstan | 23 | 13 | Azerbaijan | 8 | 13 | Bosnia and Herzegovina | 23 |
| 14 | Philippines | 22 | 13 | Botswana | 8 | 14 | Iceland | 22 |
| 15 | Bahrain | 15 | 13 | Mali | 8 | 15 | Latvia | 18 |
| 15 | Jordan | 15 | 13 | Zimbabwe | 8 | 16 | Albania | 17 |
| 17 | Kuwait | 12 | 17 | Democratic Republic of Congo | 7 | 16 | Bulgaria | 17 |
| 17 | Sri Lanka | 12 | 18 | Benin | 5 | 18 | Kosovo | 12 |
| 19 | Iraq | 11 | 18 | Burundi | 5 | 19 | Macedonia | 9 |
| 20 | Brunei | 10 | 20 | Uzbekistan | 4 | 19 | Malta | 9 |
| 20 | Rwanda | 10 | 21 | Cape Verde | 3 | 21 | Georgia | 6 |
| 22 | Kyrgyzstan | 9 | 21 | Niger | 3 | 21 | Montenegro | 6 |
| 23 | Afghanistan | 7 | 23 | Eswatini | 2 | 23 | Moldova | 3 |
| 23 | Fiji | 7 | 23 | Republic of Congo | 2 | 24 | Syria | 1 |
| 25 | Senegal | 6 | 23 | Sierra Leone | 2 | | | |
| 25 | Yemen | 6 | 26 | Togo | 1 | | | |
| 27 | Mongolia | 5 | | | | | | |
| 28 | Haiti | 3 | | | | | | |
| 29 | Bhutan | 2 | | | | | | |
| 29 | Sudan | 2 | | | | | | |
| 31 | North Korea | 1 | | | | | | |

| Cluster 4 | | | Cluster 5 | | | Cluster 6 | | |
|---|---|---|---|---|---|---|---|---|
| Rank | Country | Weight | Rank | Country | Weight | Rank | Country | Weight |
| 1 | Spain | 80 | 1 | Greece | 42 | 1 | Finland | 70 |
| 2 | Ireland | 54 | 1 | Iran | 42 | 2 | India | 60 |
| 3 | Chile | 41 | 3 | Japan | 41 | 3 | Bangladesh | 33 |
| 4 | Mexico | 40 | 4 | Egypt | 36 | 4 | Nepal | 13 |
| 5 | Peru | 34 | 5 | Oman | 30 | 4 | North Ireland | 13 |
| 6 | Colombia | 33 | 6 | Cyprus | 29 | 6 | Namibia | 3 |
| 7 | Argentina | 23 | 7 | Qatar | 24 | 7 | Grenada | 1 |
| 8 | Ecuador | 18 | 7 | Tunisia | 24 | 7 | Somalia | 1 |
| 9 | Costa Rica | 12 | 9 | Lebanon | 17 | | | |
| 10 | Venezuela | 8 | 10 | Libya | 7 | | | |
| 11 | Cuba | 7 | 11 | Cote Ivoire | 6 | | | |
| 11 | Uruguay | 7 | 11 | Palestine | 6 | | | |
| 13 | Panama | 5 | 13 | Curacao | 3 | | | |
| 14 | Nicaragua | 4 | | | | | | |
| 15 | Bolivia | 2 | | | | | | |
| 16 | Andorra | 1 | | | | | | |
| 16 | El Salvador | 1 | | | | | | |

**Table A4.** *Cont.*

| Cluster 7 | | | Cluster 8 | | | Cluster 9 | | |
|---|---|---|---|---|---|---|---|---|
| Rank | Country | Weight | Rank | Country | Weight | Rank | Country | Weight |
| 1 | France | 82 | 1 | Canada | 86 | 1 | Germany | 88 |
| 2 | Scotland | 61 | 2 | New Zealand | 53 | 2 | Netherlands | 75 |
| 3 | Israel | 26 | 3 | Jamaica | 10 | 3 | Austria | 56 |
| 4 | Monaco | 3 | 4 | Barbados | 3 | 4 | Liechtenstein | 17 |
| 5 | Algeria | 2 | 5 | Chad | 2 | 5 | Armenia | 5 |
| 6 | Madagascar | 1 | 6 | Tonga | 1 | | | |
| 6 | Paraguay | 1 | | | | | | |

| Cluster 10 | | | Cluster 11 | | | Cluster 12 | | |
|---|---|---|---|---|---|---|---|---|
| Rank | Country | Weight | Rank | Country | Weight | Rank | Country | Weight |
| 1 | Italy | 84 | 1 | England | 120 | 1 | USA | 115 |
| 2 | Portugal | 60 | 2 | Trinidad and Tobago | 6 | 2 | Guatemala | 2 |
| 3 | Norway | 45 | 3 | Mauritius | 2 | 3 | Burkina Faso | 1 |
| 4 | Mozambique | 4 | 4 | Tajikistan | 1 | 3 | Seychelles | 1 |
| 5 | San Marino | 1 | | | | | | |

| Cluster 13 | | | Cluster 14 | | | Cluster 15 | | |
|---|---|---|---|---|---|---|---|---|
| Rank | Country | Weight | Rank | Country | Weight | Rank | Country | Weight |
| 1 | Sweden | 65 | 1 | Brazil | 39 | 1 | Belgium | 59 |
| 2 | Denmark | 62 | 2 | Cambodia | 8 | 2 | Wales | 47 |
| 3 | Singapore | 43 | 3 | Belarus | 7 | 3 | Luxembourg | 17 |
| 4 | Zambia | 6 | | | | | | |

| Cluster 16 | | | Cluster 17 | | | Cluster 18 | | |
|---|---|---|---|---|---|---|---|---|
| Rank | Country | Weight | Rank | Country | Weight | Rank | Country | Weight |
| 1 | New Caledonia | 3 | 1 | Guinea | 0 | 1 | Guyana | 0 |
| 1 | Palau | 3 | | | | | | |

| Cluster 19 | | | Cluster 20 | | | Cluster 21 | | |
|---|---|---|---|---|---|---|---|---|
| Rank | Country | Weight | Rank | Country | Weight | Rank | Country | Weight |
| 1 | Laos | 0 | 1 | Papua New Guinea | 0 | 1 | Swaziland | 0 |

| Cluster 22 | | |
|---|---|---|
| Rank | Country | Weight |
| 1 | Honduras | 3 |

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
