# Peer review of "A Comparative Analysis of Social Entrepreneurship and Entrepreneurship: An Examination of International Co-Authorship Networks"

_sustainability, doi:10.3390/su152215873_

Round 1
Reviewer 1 Report
Comments and Suggestions for Authors
After analyzing the proposed article, I must congratulate the authors because the analysis of social entrepreneurship is a relevant topic in the research of any entrepreneur. In addition to congratulations for the ease of the vocabulary used as well as the readability of the article. Although I advise that some recommendations have been taken into account
In the introductory section, the research gap covered as well as the objective pursued in the research must be additionally justified, without forgetting to refer to the parts of the research. It would also be useful to make a brief mention of the state of the art so far in the introductory section.
In the literature review section, the authors have made a clear and concrete reference. However, I recommend that the literature review be more extensive to give the actor a more complete idea of the state of the art so far and to more concretely justify the research questions posed in section 2.4.
I recommend that Figure 2, as well as the tables presented here, be produced using more modern methodologies and better visualization techniques.
Once the literature review section is completed, it would be necessary to complete the discussion section by referencing these new references.
Once all these issues have been considered, it would be necessary to review the manuscript again before making a decision about its publication.
Comments on the Quality of English LanguageMinor editing of the English language is required.
Author Response
Dear Reviewers,
First, we would like to show our appreciation on your comments and suggestions. All the comments and suggestions were truly valuable to improve our paper. We have given our best to follow all your comments and suggestions in the following. We wish that our responses met your expectations and intentions. Again, we are very grateful for your support and cooperation.
Warm Regards,
|
Reviewer #1 |
The research gap covered as well as the objective pursued in the research must be additionally justified, without forgetting to refer to the parts of the research. |
Thank you for your suggestion. I agree that the research gap and objectives need more profound discussion. We have added references to support our statement and discussion from different angles to deepen our understanding to the research gap. Line116, Line 140 |
|
Reviewer #1 |
Make a brief mention of the state of art so far in the introductory section. |
Thank you for your insightful comment. We fully understand that it is important to mention the state of art in the introduction. We have added sentences to explain our state of art at the end of introduction section. Line 53 |
|
Reviewer #1 |
The literature review section should be more extensive to give the actor a more complete idea of the state of the art so far and to more concretely justify the research question posed in section 2.4.
|
Thank you for your suggestion. We truly believe that more in-depth and wide consideration on the theme can support readers to fully understand the development process of research question. We have added new discussion in the literature review section as well as in the research question section. Line 67 |
|
Reviewer #1 |
Figure 2 and the table presented should be produced using modern methodologies and better visualization techniques. |
Thank you for your suggestion. We have followed the recent study which conducted bibliometric analysis to meet the modern style of figure to improve our visualization. Line 324 |
|
Reviewer #1 |
After the literature review section to be completed, it is necessary to complete the discussion section by referencing new references.
|
Thank you for your guidance. Following your advice, we have referred to new references in the literature review section. Line 460 |
Reviewer 2 Report
Comments and Suggestions for Authors
It is a good work about the identification of the boundaries between the social entrepreneurship and the entrepreneurship through a bibliometric comparative analysis of International co-authorship network. I have nothing about the great effort the authors did to accomplish the RQ presented.
Unfortunately, the exposed limitations in terms of the disparity between articles from each field of research, the solely use of the English WoS database and the lack of empirical documents that were not published in the form of WoS academic articles, make me doubt about the relevance of the proposed submission.
I have been working both as an activist and publishing as an academic during the last 30 tears and I could agree with some of the conclusions the authors obtained. However I couldn't agree with the great quantity of quality of the research leaft aside. In fact, the main absence about the Dr Yunus research and practice on social entrepreneurship are notorious. That make me think about relevance of the proposed work in terms of the practical and successful experiences versus academic articles in the WoS database. I really would like to see this proposal with more empirical sense.
Author Response
Dear Reviewers
First, we would like to show our appreciation on your comments and suggestions. All the comments and suggestions were truly valuable to improve our paper. We have given our best to follow all your comments and suggestions in the following. We wish that our responses met your expectations and intentions. Again, we are very grateful for your support and cooperation.
Warm Regards,
|
Reviewer #2 |
Exposed limitation in terms of disparity between articles from each field of research, the solely use of English Web of Science database and the lack of empirical documents that were not published in the form of web of science articles make me doubt about the relevance of the proposed submission. |
Thank you for your comment. We fully understand your comment. Therefore, we have added some sentences in the limitation section to minimize other reader’s concern in the future. First, it is not sufficient to include publication data that author cannot fully understand. Moreover, other language might have different terms and understanding the words entrepreneurship and social entrepreneurship. Therefore, it is important for the native speakers to conduct analysis while unifying the understanding of the concept specific to different language and countries. Regarding the article-based publication, current study aimed at understanding the researcher’s interaction in the international level to minimize the gap between practical and theoretical understanding of research field. Current data analysis and results are in line with the conclusion stated and supported by the previous study which conducted qualitative analysis. Line 534, Line 538 |
|
Reviewer #2 |
The main absence about the Dr Yunus research and practice on social entrepreneurship are notorious. That make me think about relevance of the proposed work in terms of the practical and successful experiences versus academic articles in the WoS database. I really would like to see this proposal with more empirical sense. |
Thank you for your comment. It is our shortcomings that our current paper lack on discussion regarding successful practice conducted by practitioners including Dr. Yunus who is admired by the number of entrepreneurship and social entrepreneurship researchers including ourselves. Therefore, as you suggested we would like to include successful achievement created by the great predecessors in the introduction section. Line 39 |
Reviewer 3 Report
Comments and Suggestions for Authors
The study is interesting, analyzes an interesting topic, examines international co-authorship networks, and examines the relationships between authors from different countries who sign the same paper.
It would have also been interesting to know the reason for the relationships between authors. Know the motivations: friendship, kinship, common interests, topics of interest…
Author Response
Dear Reviewers,
First, we would like to show our appreciation on your comments and suggestions. All the comments and suggestions were truly valuable to improve our paper. We have given our best to follow all your comments and suggestions in the following. We wish that our responses met your expectations and intentions. Again, we are very grateful for your support and cooperation.
Warm Regards,
|
Reviewer #3 |
It would have also been interesting to know the reason for the relationships between authors. Know the motivation, friendship, kinship, common interests, topic of interest. |
Thank you for your comments and suggestions. The relationship between authors was based on whether two or more countries have developed a publication together. Moreover, the reasons why this relationship are developed can only be estimated through the trend in results which we have perceived. First is that common language, culture, and understanding can bring researchers from different country together. On the other hand, when countries share common interests in a particular topic can also bring researchers together for conducting study. Either way, without reaching out to the researchers and asking on the reason why they decided to conduct study together we would not be able to identify the reason behind them. |
Reviewer 4 Report
Comments and Suggestions for Authors
This insightful research identifies co-authorship international engagement as an indicator for distinguishing entrepreneurship from social entrepreneurship.
While the logic and coherence of the analysis are constructive, the study lacks a clear identification of the defining characteristics of social entrepreneurship. While authors recognize that the field is "new," at least about entrepreneurship, the lack of exploration of the diversity of interpretations and definitions of social entrepreneurship may limit the scope and representation of the research, especially in not-English countries or in national systems that define social enterprises in different legal terms outside and beyond "nonprofit" and "entrepreneurship. " Think, for example, Latin American countries and their emphasis on social and solidarity economy that may or may not be represented by "social entrepreneurship" definitions here contemplated.
Even cultures play a role in the definition of SE. "Definitions of SE are often vague, covering a wide variety of activities and representing different models worldwide. The multiplicity of actors involved in innovative and social activities, as well as the variety of motives that lie upon their adoption – from profit-driven to voluntarily to philanthropic not-for-profit – often leads to a misunderstanding about the concept."
https://www.sciencedirect.com/science/article/pii/S0148296320300126?casa_token=3M4lcA-D62cAAAAA:8WvtTDCvYpi7FzQbX8Ah9aNmlxLphuWLG0zed5FgKnvi_dBnMzH140tQy20_AIIUmsL2EF1_mw
Indeed, social entrepreneurship is a “simple term with a complex range of meanings” and this research does not give sufficient attention to these complexities. https://link.springer.com/article/10.1007/s11187-011-9398-4
This lack of understanding of SE, is exemplified in the first sentence of the paper:
"Social entrepreneurship is a nonprofit organization which applies entrepreneurial strategies to sustain themselves financially," which appears to reference Gupta, P.; Chauhan, S.; Paul, J.; Jaiswal, M.P. Social Entrepreneurship Research: A Review and Future Research 568 Agenda. J. Bus. Res. 2020, 113, 209–229, doi:10.1016/j.jbusres.2020.03.032. In reality, "social entrepreneurship" is a field of study and practice... and it is not a "nonprofit organization that...". This is incorrect.
Eventually, it could be said: "Social entrepreneurship is the field in which entrepreneurs tailor their activities to be directly tied with the ultimate goal of creating social value." https://timreview.ca/sites/default/files/article_PDF/Saifan_TIMReview_February2012_2.pdf
This paper and study could benefit from reviewing and integrating a more comprehensive analysis of the definition of SE, considering its diversity in the definitions and practices internationally. https://www.scirp.org/journal/paperinformation.aspx?paperid=127680
Comments on the Quality of English Language
Some sentences are not so clearly written and could be edited.
Author Response
Dear Reviewers,
First, we would like to show our appreciation on your comments and suggestions. All the comments and suggestions were truly valuable to improve our paper. We have given our best to follow all your comments and suggestions in the following. We wish that our responses met your expectations and intentions. Again, we are very grateful for your support and cooperation.
Warm Regards,
|
Reviewer #4 |
The study lacks a clear identification of defining characteristics of social entrepreneurship. |
Thank you for your suggestion. We agree with your comment. As suggested, we would like to add a sentence which explain the current study’s social entrepreneurship definition in the introduction section. Line 78 |
|
Reviewer #4 |
The lack of exploration of the diversity of interpretation and definition of social entrepreneurship may limit the scope and representation of the research, especially in non-English countries or in national systems that define social enterprises in different legal terms outside and beyond “nonprofit” and entrepreneurship. |
Thank you for your comment. We fully understand and agree with your opinion. It is important to consider the different interpretation of social entrepreneurship, especially in non-English speaking countries as well as to consider different national system which cause the difference identification on social entrepreneurship. Since, our focus is to identify the difference in the research field of social entrepreneurship and entrepreneurship from international co-authorship. It was considered not sufficient for us to include non-English written articles which we did not fully understand. However, in the further study, we would like to focus on each country with their different definition of social entrepreneurship as well as its conceptualization influenced by culture, language, and national system while including entrepreneurship researcher in other nation especially in Spain and China which had relatively higher ratio of article written in the field of social entrepreneurship after English. We have added a sentence which explains our decision regarding language limitation and publication-based selection of articles. Line 534, Line 539 |
|
Reviewer #4 |
Social entrepreneurship is a simple term with a complex range of meaning and this research does not give sufficient attention to these complexities. |
Thank you for your comment. We fully understand and agree with your point. As suggested, we would like to add more discussion to the definition of social entrepreneurship which correspond to Table 1. Line 67 |
|
Reviewer #4 |
The lack of understanding of SE, is exemplified in the first sentence of the paper. “social entrepreneurship is a nonprofit organization which applies entrepreneurial strategies to sustain themselves financially” Social entrepreneurship research: A review and future research. doi: 10. 1016/j.jbusres.2020.03.032. |
Thank you for pointing out on this matter. This is completely our fault in creating the sentence which cause wrong identification of what social entrepreneurship is. Moreover, we do not perceive social entrepreneurship equals to non-for-profit organization. Therefore, we have changed the sentence as suggested. Again, thank you for your comment. Line 34 |
|
Reviewer #4 |
Eventually, it could be said: "Social entrepreneurship is the field in which entrepreneurs tailor their activities to be directly tied with the ultimate goal of creating social value." https://timreview.ca/sites/default/files/article_PDF/Saifan_TIMReview_February2012_2.pdf |
Thank you for bringing this recent citation to my attention. I’ve added this sentence in the first sentence of introduction section to explain what social entrepreneurship is as an introduction. Line 34 |
|
Reviewer #4 |
This paper and study could benefit from reviewing and integrating a more comprehensive analysis of the definition of SE, considering its diversity in the definitions and practices. |
Thank you for your suggestion. We really appreciate your sincere guidance. Current study was developed with the goal of identifying the definition of social entrepreneurship through analyzing from regional aspects. Therefore, we have reviewed some study to incorporate for analyzing the definition of social entrepreneurship from different aspects. We’ve added more in-depth explanation of different types of definition of social entrepreneurship (Line 67) and practical example of successful social entrepreneurship as well (Line 39).
|
Reviewer 5 Report
Comments and Suggestions for Authors
- introduction is very short. the authors need to add value or study novelty. They also need to quickly summarize the literature review on the topic. Finally, they should add the paper design.
- research question should be placed rather in the introduction. the authors need to add the hypotheses of the study at the end of the literature eview section.
- why data collection is stopped at 2021 and not 2023?
- other parts are fine
Author Response
Dear Reviewers,
First, we would like to show our appreciation on your comments and suggestions. All the comments and suggestions were truly valuable to improve our paper. We have given our best to follow all your comments and suggestions in the following. We wish that our responses met your expectations and intentions. Again, we are very grateful for your support and cooperation.
Warm Regards,
|
Reviewer #5 |
Introduction is very short. The authors need to add value or study novelty. They also need to quickly summarize the literature on the topic. |
Thank you for your comment. We agree with your comments regarding the volume of introduction section. We’ve added some sentence to explain the novelty of this paper. Also, we will add the quick summarize to the literature. Line 53
|
|
Reviewer #5 |
They should add the paper design. |
Thank you for your comment. We have developed paper design. Please refer to the attachment files. |
|
Reviewer #5 |
Research question should be placed rather in the introduction. |
Thank you for your suggestion. We consider the literature review section as the part of introduction section. Therefore, as suggested, we would like to place the research question at the end of introduction section. Line 157 |
|
Reviewer #5 |
The author needs to add the hypotheses of the study at the end of the literature review section. |
Thank you for your suggestion. We have placed our literature review while following the MDPI’s instruction for authors. Regarding research question which corresponds to hypothesis is placed in the introduction section. Line 157 |
|
Reviewer #5 |
Why data collection is stopped at 2021 and not 2023? |
Thank you for your question. The reason is that the number of publications fluctuate for one year, therefore, it is challenging to define that the year of publication after a year. Therefore, setting the year to 2021 allows us to collect all the publication including early publication to late publication to be regarded as 2021. When conducting bibliometric analysis, the fiscal year is generally set one year shorter to prevent contamination caused by variations in the timing of papers. |
Round 2
Reviewer 1 Report
Comments and Suggestions for Authors
Dear Editor,
after reviewing the manuscript, I recommend its publication in its current format after being modified in accordance with the suggestions made.
Kind regards
Reviewer 2 Report
Comments and Suggestions for Authors
Thank you so much for the improvements in your work, I am happy to help you to do that,
Reviewer 4 Report
Comments and Suggestions for Authors
Thank you for making the adjustments and improving the literature review.
Reviewer 5 Report
Comments and Suggestions for Authors
Dear authors,
Thank you for your effort to revise the paper based on the reviewers' comments.